# Head Information Bottleneck: An Evaluation Method for Transformer Head Contributions in Speech Tasks

## Abstract

Multi-head attention mechanisms have been widely applied in speech pre-training. However, their roles and effectiveness in various downstream tasks have not been fully studied. Different attention heads may exhibit varying degrees of importance in different downstream tasks. We noticed that the attention allocation in the attention mechanism is similar to the information bottleneck, aiming to highlight the parts important to the task. Therefore, we introduced the information bottleneck into multi-head attention to estimate the degree of mutual information contained in each attention head's output about the input and forced it to focus on useful information. Additionally, we proposed a method to measure the contribution of attention heads in tasks. We also pruned the model heads according to their contributions, providing an interpretable direction for model pruning. Notably, our method can maintain an accuracy of 83.36% on the Keyword Spotting (KS) task while pruning 40% of the heads.

## 1 Introduction

Self-supervised learning (SSL) pre-trained models, such as wav2vec 2.0 (Baevski et al., 2020), Hu-BERT (Hsu et al., 2021), and WavLM (Chen et al., 2022), have become leading technologies in speech-related applications, outperforming traditional systems like HMM-GMM (Rabiner, 1989). However, the complexity of end-to-end ASR models makes them less transparent and hard to interpret, sometimes leading to unreliable outputs (Weidinger et al., 2021). Additionally, the large number of parameters in these models presents difficulties in pruning and compression, as not all parameters are equally beneficial for various tasks (Chen et al., 2020; Hou et al., 2020). Therefore, it's essential to enhance the explainability of these models to understand their functioning better.

Explainability is the ability to make model actions understandable to humans (Doshi-Velez & Kim, 2017; Du et al., 2019). Many studies offer layer-level insights (Pasad et al., 2021; Yang et al., 2021), like SUPERB, which assigns weights to layers for tasks to show each layer's role. However, such exploration is not detailed enough, and with the introduction of attribution methods (Selvaraju et al., 2017; Zeiler & Fergus, 2014; Smilkov et al., 2017), researchers have begun to interpret the attention mechanism within layers. These attribution methods explain model behavior by assigning relevance scores to each input variable (Yang et al., 2020; Shim et al., 2021; Zhmoginov et al., 2021).

We propose a novel attribution method for assessing the contribution of attention heads in Transformers. We introduce the concept of the information bottleneck in the attention mechanism, utilizing a newly initialized weight matrix and estimating the upper bound of mutual information from input to output based on variational approximation. The resulting head information bottleneck attention matrix is compared with the model's original attention matrix for similarity calculation to obtain the final head contribution parameters. For reproducibility, we share our source code and provide an easy-to-use implementation. We name our method Head Information Bottleneck Attribution (HIB). To explore the effectiveness and explainability of HIB, we conducted tests and validations in multiple speech downstream tasks. In summary, the main contributions of this paper are as follows:

1. We introduce the concept of the information bottleneck in the attention mechanism to estimate the degree of mutual information between output and input in each attention head.

The information bottleneck theory provides an effective guarantee to compel the model to focus on useful information from the input.

2. We propose a method to calculate the contribution of Transformer attention heads based on the head information bottleneck attention matrix, characterizing the importance of attention heads in the model.

3. We provide an interpretable direction for model pruning, enabling targeted deletion of attention heads in the model.

## 2 BACKGROUND

### 2.1 SELF-ATTENTION AND MULTI-HEAD ATTENTION

In recent years, speech research has revealed that utilizing SSL pre-trained speech models as encoders for fine-tuning can significantly improve performance in various downstream speech tasks (Baevski et al., 2022; Ling & Liu, 2020). These models employ innovative methods to construct powerful encoders that provide high-quality audio contextual representations for decoding. The network architectures adopted by these models are primarily based on Transformer (Vaswani et al., 2017) or Conformer (Gulati et al., 2020) structures featuring attention mechanisms. Within self-attention, each element of the input sequence $X$ is linearly projected into three distinct spaces, forming key, value, and query representations:

$$Q = XW_Q \quad K = XW_K \quad V = XW_V \tag{1}$$

Where $W_Q$, $W_K$, and $W_V$ are learnable weight matrices. The attention mechanism computes scores between the query and all keys for each sequence element, typically normalized by a constant for gradient stability:

$$A = \text{softmax}(\frac{Q \cdot K^T}{\sqrt{d_k}}) \quad O = A \cdot V \tag{2}$$

Where $d_k$ denotes the dimension of the key. During the computation of attention scores, multi-head attention is commonly employed, enabling the model to capture diverse information across multiple representational subspaces. By capturing these diverse dependencies, the representational capacity of the model is enhanced. The mathematical representation of multi-head attention is as follows:

$$Q_i = XW_{Qi} \quad K_i = XW_{Ki} \quad V_i = XW_{Vi} \tag{3}$$

Where $W_{Qi}$, $W_{Ki}$, and $W_{Vi}$ are the weight matrices for the $i^{th}$ head. The corresponding attention scores are computed with the respective value to yield the output for each attention head. These outputs are concatenated to form the final output:

$$A_i = \text{softmax}(\frac{Q_i \cdot K_i^T}{\sqrt{d_k}}) \quad O_i = A_i \cdot V_i \quad O = \text{Concat}(O_1, \cdots, O_i, \cdots, O_n) \tag{4}$$

Where $n$ represents the number of attention heads.

### 2.2 INFORMATION BOTTLENECK

The Information Bottleneck(IB) theory (Tishby et al., 2000) furnishes a theoretical framework that delineates the limitations of accessible information. Conventionally, the prediction of a label $Y$ is contingent upon harnessing the information available from input $X$. The innovation of the IB theory lies in introducing an intermediary random variable $Z$, thus reconfiguring the predictive sequence into $X \to Z \to Y$. The core objective of the IB theory is to maximize the mutual information between the random variable $Z$ and the label $Y$ while simultaneously minimizing the mutual information between $Z$ and the input $X$. The formula succinctly captures this dual objective:

$$\max I[Z; Y] - \beta I[Z; X] \tag{5}$$

Where $I[Z; Y]$ and $I[Z; X]$ represent the mutual information, and $\beta$ is used to control the balance between making accurate label predictions and using minimal information from $X$. Increasing $\beta$ will utilize less information from $X$. This formulation creates a funnel-like passage of information,

with $Z$ acting as the bottleneck in the middle. Yet, the computation of mutual information usually demands detailed knowledge of the random variables' distribution, an often impractical condition in numerous contexts. To address this, variational techniques have been devised to approximate mutual information(Alemi et al., 2016). Within these variational frameworks, the distribution of $Z$ is often posited as a standard normal distribution, thereby streamlining the complexity of calculations.

## 2.3 PREVIOUS WORK

Attribution research is continually evolving. Gradient Maps (Baehrens et al., 2010), or local gradient explanation vectors, show how classifiers decide by calculating gradients of outputs over inputs. Integrated Gradient (Sundararajan et al., 2017) introduces Sensitivity and Implementation Invariance. SmoothGrad (Smilkov et al., 2017) adds noise to images to assess pixel sensitivity. Both methods analyze gradients at inputs and the path to anchor points. Deep Taylor Decomposition (DTD) (Montavon et al., 2017) redistributes output contributions back to inputs, while PatternAttribution (Kindermans et al., 2018) separates signal and interference, using DTD to discern feature contributions without backpropagating interference. Occlusion (Zeiler & Fergus, 2014) and blurred blocks (Greydanus et al., 2018) gauge importance by zeroing or blurring parts of an image and noting accuracy drops. Information bottleneck approaches (Achille & Soatto, 2018; Alemi et al., 2016; Zhmoginov et al., 2021; Taghanaki et al., 2019) limit information flow in networks to predict the importance of image regions, with variational approximation (Schulz et al., 2020) bounding mutual information for attribution.

Transformer architecture advancements have spurred deeper inquiry into the attribution of attention mechanisms. Adjusting attention weights manually can reveal their influence on the model (Jaunet et al., 2021). Deeper dives involve visualizing attention flows across heads and layers for a more nuanced model understanding (Park et al., 2019; Hoover et al., 2019; Yeh et al., 2023), tracking how information morphs through the model and comparing different models' training phases (DeRose et al., 2020). However, raw attention visualization alone may not fully explain model predictions, prompting the creation of advanced methods like Grad-SAM (Barkan et al., 2021), which uses gradients to produce self-attention heatmaps, and AttAttr (Hao et al., 2021), which computes gradient sums from a zero attention matrix for self-attention attribution scores.

Inspired by the information bottleneck attribution method, we have constructed a novel information bottleneck acquisition method for the attention mechanism of Transformers. It is entirely different from previous work. We do not make any noise or mask changes to the already trained attention heads. Instead, we completely reconstruct an attention score to compress the information of values to obtain an information bottleneck. We judge each attention head's contribution to different downstream tasks using the information bottleneck results. To our knowledge, this is the first time the information bottleneck theory has been used to analyze the roles of different attention heads in a model.

## 3 HEAD INFORMATION BOTTLENECK

Our innovation is devising an attribution method for attention heads within a trained network. This method is integrated into the forward propagation process of such networks. We have innovatively replaced the traditional attention components of the attention mechanism with an information bottleneck. The overall structure is illustrated in Fig.1. In executing this method, we impose an information bottleneck on the $V$(value) portion of the attention mechanism while concurrently maximizing the model's original goals. Throughout this process, we ensure that the remaining original parameters of the model remain intact.

### 3.1 METHOD

We recognize some commonalities between the information bottleneck and attention mechanisms. In the attention mechanism, the role of attention is to recombine vectors within the $V$ to assign greater weights to the valuable information in the $V$. The IB theory aims to predict outcomes by utilizing as little information from the original inputs as possible. When we regard $V$ as the original input, the two strategies become consistent, seeking to utilize valuable information within $V$. Therefore, we apply the IB theory to the multi-head attention mechanism to leverage minimal information from the $V$ to capture the genuinely beneficial elements of the task.

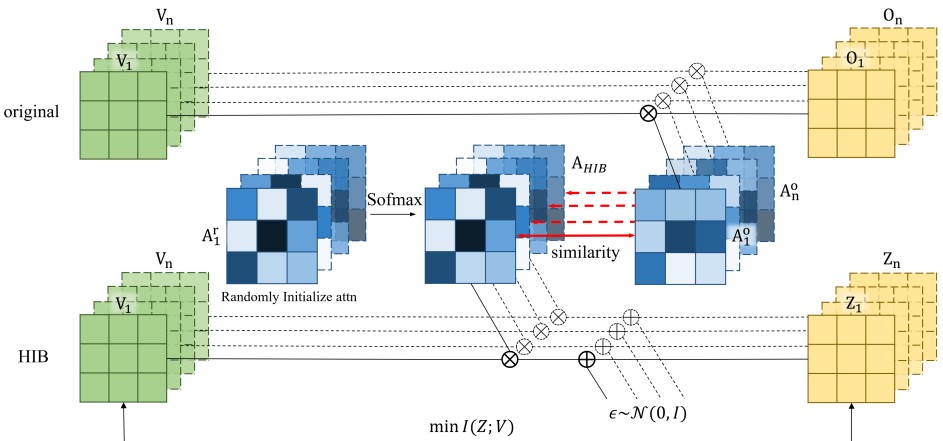

Figure 1: Head Information Bottleneck structure diagram

Specifically, we randomly initialize a weight matrix $A_r \in \mathbb{R}^{L \times L}$, which is designated to take the place of the attention matrix within the attention mechanism. This matrix is employed to apply weights to the matrix $V \in \mathbb{R}^{L \times d}$ in the attention scheme. Additionally, we incorporate noise $\epsilon \in \mathbb{R}^{L \times d}$, thereby constructing the intermediate stochastic vector $Z \in \mathbb{R}^{L \times d}$, where $L$ represents the length of the sequence and $d$ corresponds to the dimension of each vector in the attention heads. The stochastic variable $Z$ is in alignment with the random variable $Z$ as referred to in Eq.5.

Eq.6 specifically illustrates how to generate the intermediate random variable $z_i \in \mathbb{R}^{1 \times d}$ for the $i$-th row from the matrix $V$.

$$z_i = a_i^r \cdot V + \lambda \epsilon_i \quad \epsilon_i \sim \mathcal{N}(0, I_d) \tag{6}$$

Where $a_i^r \in \mathbb{R}^{1 \times L}$ denotes the $i$-th row of the weight matrix, and $\epsilon_i \in \mathbb{R}^{1 \times d}$ represents a random variable from the standard multivariate normal distribution.

Noise is incorporated for a twofold purpose: firstly, it injects randomness into the variable $Z$, thereby permitting the computation of its mutual information with $V$; secondly, the noise serves to modulate the flow of pertinent information.

Since the information bottleneck is introduced within the attention mechanism, the original input $X$ in Eq.5 is replaced by the matrix $V$ from the attention mechanism. Consequently, Eq.5 is transformed as follows:

$$\max I[Z; Y] - \beta I[Z; V] \tag{7}$$

We aim to constrain the utilization of information from the $V$ through the information bottleneck. Hence we need to minimize $I(Z; V)$ whose general formula is given by:

$$I[Z; V] = \mathbb{E}_V[D_{KL}[P(Z|V)||P(Z)]] \tag{8}$$

To apply the information bottleneck for limiting the use of information from the $V$, it's necessary to know the actual distribution of $Z$, but the true expression for $P(Z)$ is beyond our reach. Therefore, we turn to variational approximation for estimation. After undergoing linear mapping, We assume that the features of the matrix $V$ are Gaussian distributed and independent of one another. This is a reasonable assumption since features often exhibit Gaussian distribution following linear or convolutional layers (Klambauer et al., 2017; Schulz et al., 2020).

Given our previous definition of $z_i$ as $z_i = a_i^r \cdot V + \lambda \epsilon_i$, which represents the linear combination of the row vectors in $V$ followed by the addition of noise $\epsilon_i$, it follows that $z_i$ also adheres to a Gaussian distribution. Consequently, we can infer that $Q(Z)$ maintains a Gaussian distribution. For the sake of computational simplicity, we assume that the features $z_i$ within $Q(Z)$ are distributed according to a standard Gaussian distribution. This assumption aligns with the common practices in variational methods (Alemi et al., 2016). However, the independence assumption usually does not hold in practice, which only results in overestimating mutual information. Substituting $Q(Z)$ into

Eq.8, we get:

$$I(Z;V) = \mathbb{E}_V[D_{KL}[P(Z|V)||Q(Z)]] - D_{KL}[P(Z)||Q(Z)] \tag{9}$$

Since the divergence in the second term is non-negative, the lower bound of mutual information $I(Z;V)$ is $\mathbb{E}_V[D_{KL}[P(Z|V)||Q(Z)]]$. For detailed derivation, refer to Appendix A.

The general formula for KL divergence is:

$$D_{KL}[P||Q] = \int p(x) \log \frac{p(x)}{q(x)} dx \tag{10}$$

For the scenario where both $P(Z|V)$ and $Q(Z)$ are multivariate normal distributions, the formula for the KL divergence can be expressed as:

$$D_{KL}[P||Q] = \frac{1}{2}\left[\text{tr}(\Sigma_Q^{-1}\Sigma_P) + (\mu_P - \mu_Q)^T\Sigma_Q^{-1}(\mu_P - \mu_Q) - d + \log\frac{\det\Sigma_Q}{\det\Sigma_P}\right] \tag{11}$$

Here, $\mu_Q$ and $\mu_P$ are the means of the two distributions, $\Sigma_Q$ and $\Sigma_P$ are the covariance matrices, $d$ is the dimension, $\text{tr}[\cdot]$ is the trace of a matrix, and $\det[\cdot]$ is the determinant of a matrix. In line with the preceding hypothesis, $V$ is constituted by $L$ vectors, each conforming to a $d$-dimensional multivariate normal distribution. We use statistical data to scale it to a standard multivariate normal distribution for ease of computation. The expectation and covariance matrix for the distribution $P(z_i|V)$ can then be extrapolated from $z_i = a_i^T \cdot V + \lambda\epsilon_i$, with the calculations presented as follows:

$$\mathbb{E}[z_i] = a_i\mathbb{E}[V] + \mathbb{E}[n] = a_i \cdot 0 + 0 = 0 \tag{12}$$

$$\text{Cov}[z_i] = a_i\text{Cov}[V]a_i^T + \text{Cov}[n] = a_iI_da_i^T + I_d = a_ia_i^T + I_d \tag{13}$$

Moreover, since the $z_i$ are independent of each other, the KL divergence $D_{KL}[P(Z|V)||Q(Z)]$ can be decomposed as the sum of the individual divergences:

$$D_{KL}[P(Z|V)||Q(Z)] = \sum_{i=1}^{L} D_{KL}[P(z_i|V)||Q(z_i)] \tag{14}$$

This allows us to calculate the KL divergence for each $z_i$ separately. By substituting the obtained means and variances into Eq.11, we can derive:

$$\begin{aligned}
D_{KL}[P(z_i|V)||Q(z_i)] &= \frac{1}{2}\left[\text{tr}(I_d(a_ia_i^T + \lambda^2I_d)) - d + \log\frac{\det I_d}{\det(a_ia_i^T + \lambda^2I_d)}\right] \\
&= \frac{1}{2}\left[\sum_{j=1}^{L} a_{ij}^2 + \lambda^2 d - d - \log(||a_i||^2 + \lambda^2) - 2(d-1)\log\lambda\right]
\end{aligned} \tag{15}$$

The final information bottleneck loss is:

$$\mathcal{L}_{IB} = \frac{1}{2}\sum_{i=1}^{L}\left[\sum_{j=1}^{L} a_{ij}^2 + \lambda^2 d - d - \log(||a_i||^2 + \lambda^2) - 2(d-1)\log\lambda\right] \tag{16}$$

To maintain a high accuracy of the model, the original loss of the model also needs to be introduced, so the total loss is:

$$\mathcal{L} = \mathcal{L}_{model} + \beta\mathcal{L}_{IB} \tag{17}$$

Here, the parameter $\beta$ controls the relative importance of the two objectives. When $\beta$ is smaller, more information flows; conversely, less information flows when $\beta$ is larger. We apply the softmax function to the trained random attention matrix to obtain the head information rating attention matrix $A_{HIB}$. For more details on the choice of the softmax function, please refer to Appendix B.

## 3.2 HEAD CONTRIBUTION

Given that we maintained the constancy of other parameters during the attribution process, the trained attention head matrix $A_{HIB}$ represents the minimal usage of information from $V$. Hence, a high similarity between $A_{HIB}$ and the original attention matrix $A_O$ would suggest that the attention head was already utilizing information optimally. In contrast, a lower similarity implies that the attention head $A_O$ contains a substantial amount of redundant information. Consequently, we propose that gauging the similarity between $A_{HIB}$ and $A_O$ can reflect the significance of the attention head to a certain extent. We define the similarity between the two as the head contribution characteristic parameter $C_h$, where the similarity algorithm for $C_h$ is derived by calculating the Pearson correlation coefficient between the two attention matrices.

## 4 EVALUATION

Inspired by the Occlusion method (Zeiler & Fergus, 2014), we assessed the effect of selectively disabling different attention heads on the total accuracy. We used the Librispeech-clean-test dataset for this experiment and conducted tests with a pre-trained wav2vec2.0 model. We investigated the role of individual attention heads by nullifying each one's matrix, effectively setting it to zero, and observed the resulting changes in model accuracy. The accuracy here is calculated using one minus the Word Error Rate (WER), quantifying the model's performance. The result is illustrated in Fig 2, where the color representation at each head's location signifies the shift in accuracy: shades of red indicate a reduction in accuracy, whereas shades of green denote an unexpected improvement in overall accuracy upon deactivating the respective head. We aim to obtain similar head contribution information through the proposed information bottleneck method, and this section will evaluate the effectiveness of the head contribution parameter $C_h$ that we have introduced.

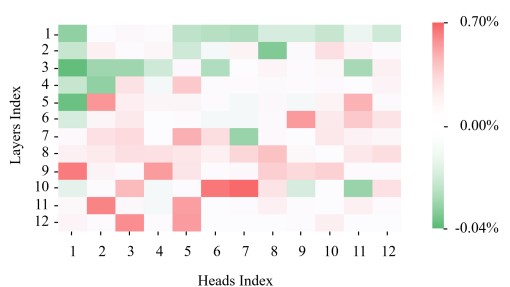

Figure 2: Impact on the overall error rate by individually masking each attention head

### 4.1 EXPERIMENTAL SETUP

We have selected the wav2vec 2.0 model, widely used in audio processing, as our primary subject of analysis. This model is fundamentally structured around a Transformer encoder and utilizes pre-trained weights provided by Huggingface. We employ the base structure of wav2vec 2.0, which comprises 12 encoder layers, each containing 12 attention heads. The dimension of the hidden layer is 768. This section exemplifies the Automatic Speech Recognition (ASR) task to analyze the proposed HIB method.

We adopt CTC-Loss on the Librispeech dataset (Panayotov et al., 2015) and fine-tune it according to the methods in Speech processing Universal PERformance Benchmark (SUPERB) (Yang et al., 2021). However, unlike the original approach, we do not fix the pre-trained parts of the wav2vec 2.0 model. Given that our computations are performed on single samples using already fine-tuned models, our optimization parameters solely consist of the random attention matrix $A_r$. The specific HIB hyperparameters are detailed in Appendix C. The trained attention matrix $A_r$, after being processed by the Softmax function, yields the final information bottleneck attention head matrix $A_{HIB}$. This matrix illustrates the distribution of attention required to complete the current task while utilizing the minimum value information.

Our optimization objective for HIB is $\mathcal{L}_{model} + \beta\mathcal{L}_{IB}$. Generally, the information bottleneck loss $\mathcal{L}_{IB}$ is significantly larger than the original model loss $\mathcal{L}_{model}$ as it aggregates the losses of all frames $L$ and attention heads $H$. To balance their relationship, we average the information bottleneck loss $\mathcal{L}_{IB}$ over frames and attention heads.

In Fig.3, we exhibit the same information bottleneck attention head matrix $A_{HIB}$ under different $\beta$ values to demonstrate the impact of $\beta$. The display here presents the situation corresponding to the

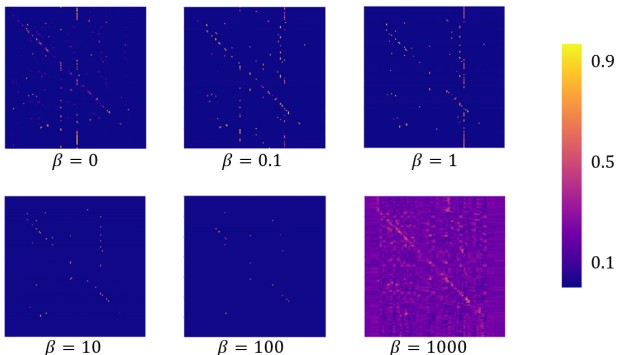

Figure 3: The impact of $\beta$ on the $A_{HIB}$.

**first attention head in the final layer.** As $\beta$ gradually increases, it implies a minimized utilization of information. When $\beta = 100$, the higher scoring regions in $A_{HIB}$ are already concentrated in a few locations, while when $\beta = 0$, indicating no restriction on mutual information, the model tends to utilize more original information. When $\beta = 1000$, due to its excessive value, it has already interfered with the normal training of the model, and at this point, the model can no longer produce the transcribed text correctly. This demonstrates that the proposed information bottleneck loss $\mathcal{L}_{IB}$ indeed serves to limit mutual information. We have chosen a balanced setting of $\beta = 10$, allowing $A_{HIB}$ to retain useful information without overly compressing it.

## 4.2 QUALITATIVE ASSESSMENT

We have replicated the AttAttr method for comparison and have adopted the settings presented in its original paper. The results are illustrated in Fig.4, where we have selected the second and third heads of the final layer in the ASR task as examples. Based on the experimental data from Fig.2, the overall impact on the error rate after these two heads were masked was 0.01% and 0.51%, respectively, representing two heads with relatively minor and major effects on the ASR task.

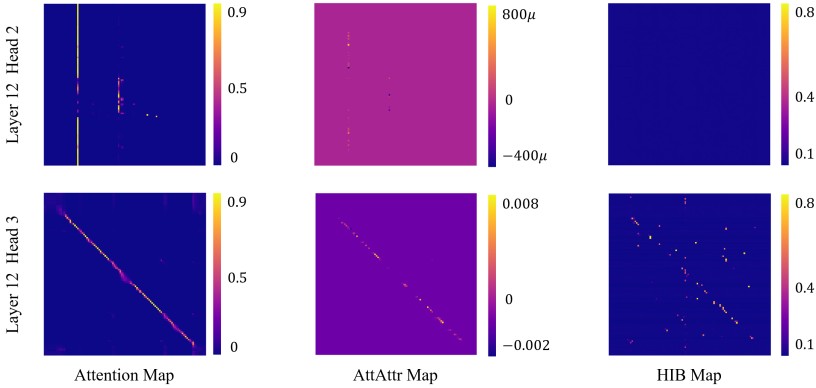

Figure 4: Different attention head attribution methods

From Fig.4, it can be observed that both HIB and AttAttr can accurately identify the active regions in the third head (second row in the figure). In our method, some discrete points are outside the diagonal due to our approach of regenerating the attention matrix without utilizing the existing attention. This leads to the possibility of some interference items. In contrast, the gradient-based AttAttr calculates based on the original attention matrix, thereby maintaining better consistency with the normal attention mechanism. This observation is also reflected in the second head (depicted in the first row of the figure), where despite its expected negligible impact on the ASR task, AttAttr still suggests the presence of relevance along the vertical lines. Conversely, our approach effectively shows that this head has little influence on the task.

### 4.3 ATTENTION HEAD PRUNING

Simultaneously, we compare the accuracy of the two algorithms in identifying key heads in each layer. AttAttr follows the method in the original paper, measuring the importance of the head through Eq.18.

$$I_h = E_x[\max(Attr_h(A))] \tag{18}$$

The term $Attr_h(A)$ is the attention attribution score proposed by the authors based on integrated gradients. It sums the gradients at points along the straight-line path from a zero attention matrix to the original attention weights $A$, with sufficiently small intervals. The expression $\max(Attrh(A))$ represents the maximum attribution value of the $h$-th attention head. The notation $E_x$ signifies taking the average over the test dataset (Hao et al., 2021).

We configured a head pruning experiment to assess the precision of head contribution measures. We ranked the heads in each layer from lowest to highest based on their contribution scores $C_h$ (from our method) and $I_h$ (from Attr's metric), sequentially masking those with lower contribution scores. Subsequently, we calculated the final accuracy. To explore whether the impact of masking individual heads could act as a measure of their contribution, we implemented a baseline method. This involved obscuring each attention head and calculating the impact on accuracy, similar to what we demonstrated in Fig2. The numerical changes in accuracy are used as the scores for head attention. We randomly extracted 200 samples from the Librispeech train-100 dataset for training. After the training was completed, we conducted tests on the test-clean dataset.

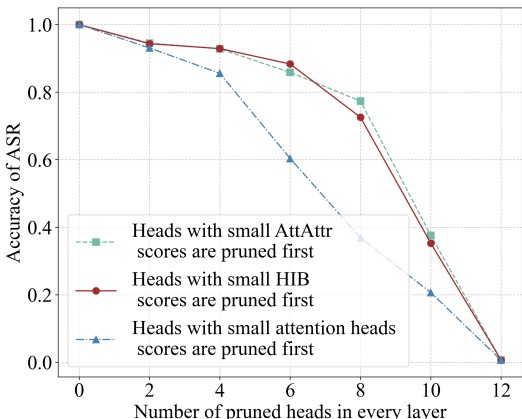

Figure 5: Evaluating the effectiveness of HIB through head pruning

The findings, shown in Figure 5, illustrate that our method retains 88% accuracy even after pruning half of the heads per layer, outperforming the baseline when the same number of heads are masked. This indicates that our method better reflects the contribution of attention heads and suggests that measuring the contribution of a head to the whole by masking it alone is not accurate. Our results are on par with those from AttAttr, with HIB pinpointing less critical heads more effectively at lower pruning levels, thus keeping model performance higher post-pruning. Only when more than eight heads are pruned does HIB begin to lag slightly behind AttAttr, confirming the efficacy of using the information bottleneck to infer attention head contributions.

### 4.4 SPEECH TASK PRUNING

We conducted head pruning tests in three speech tasks: Speaker Identification (SID), Keyword Spotting (KS), and Intent Classification (IC). We fine-tuned the pre-trained wav2vec 2.0 base model separately for these three tasks. The SID task was fine-tuned on Voxceleb1 (Nagrani et al., 2017), where the speakers are located in the same predefined set during training and testing. The KS task was fine-tuned on SpeechCommands (Warden, 2018), which includes ten common keywords and two special labels representing silence and unknown categories. The IC task was conducted on Fluent Speech Commands (Lugosch et al., 2019), where each utterance includes three intent labels: action, object, and location. The evaluation metric for these three tasks is accuracy.

All three tasks belong to sequence classification tasks. We added two linear layers to the pre-trained wav2vec 2.0 model for sequence classification. The learning rate for the wav2vec 2.0 model part of each task was set to 1e-5, and the learning rate for the linear layers used for sequence classification was set to 1e-3. We used the Adam optimizer and trained each task for 15 epochs. The final fine-tuned models achieved accuracies of 97.91%, 96.39%, and 98.71% on the three tasks, respectively.

We devised two pruning approaches for neural network optimization. The first method involves trimming the least contributive attention heads from each layer, while the second method entails a

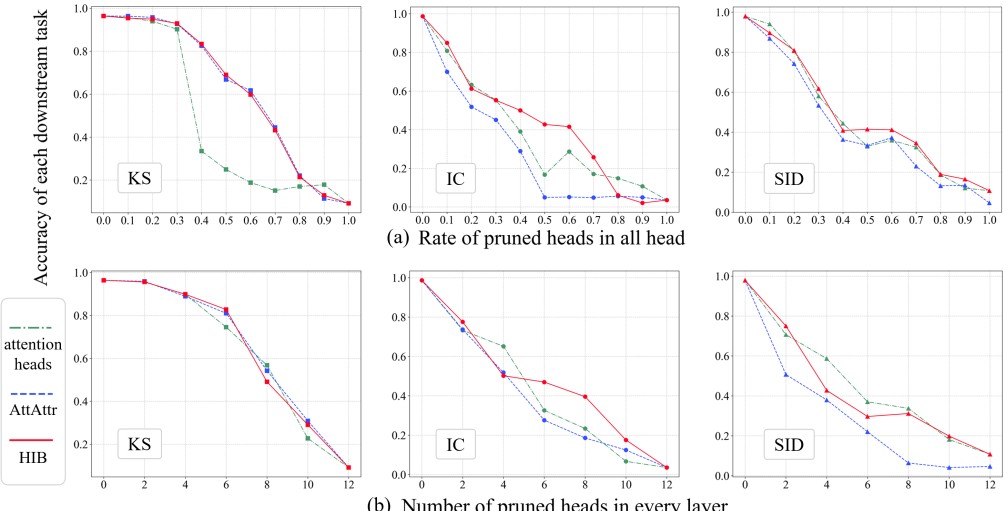

Figure 6: Perform head pruning for different speech tasks. (a) Prune each layer, starting from those with low contributions. (b) Prune all heads, starting from those with low contributions.

global ranking of attention head contributions, with pruning executed in ascending order of significance. We compared these approaches against three attribution methods—AttAttr, attention heads, and HIB. The outcomes of our comparative analysis are depicted in the accompanying Fig.6.

Our method performed well on the Keyword Spotting task, like AttAttr, retaining 82.76% accuracy even after cutting half of the heads in each layer. It also kept 83.36% accuracy after globally pruning 40% of the heads. While it did better than AttAttr on two other speech tasks, the results were not as good as expected. Global pruning was more effective than pruning by layers, suggesting that certain layers have more impact, consistent with prior WavLM research. A close look at individual head scores from AttAttr and HIB in the IC and SID tasks showed little difference, implying that the distinction between heads in these tasks is slight, potentially affecting the accuracy of our head contribution measurements.

## 5 CONCLUSION

We propose a new attribution method, HIB, for evaluating attention head contribution. It applies an information bottleneck to the attention mechanism's values, limiting their use and producing an attention matrix that minimizes value usage. This matrix is then compared to the original to evaluate each head's contribution. HIB can be used with any attention-based model, leveraging information bottleneck theory to ensure the model focuses on valuable input information. This is a significant advantage over other attention head-based attribution methods that necessitate retraining from scratch and cannot evaluate pre-trained models. Consequently, our approach substantially reduces the cost required to assess models.

In the experiment of pruning according to the head contribution, our method is similar to the previously proposed AttAttr method based on gradient backpropagation and is better than the baseline based on attention score, which verifies the effectiveness of our proposed method. At the same time, we also tested three downstream tasks: SID, KS, and IC. HIB can achieve an accuracy of 83.36% on the KS task while pruning 40% of the heads.

HIB has a certain basis in information theory, which guarantees the credibility of the method. To our knowledge, this is the first time that information bottlenecks and attention mechanisms have been linked. We have demonstrated the feasibility of this idea through experiments, and we hope that this will help deepen the understanding of neural networks and further strengthen the trust of deep learning models in sensitive application fields.

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

## A    DERIVATION OF THE UPPER BOUND OF MUTUAL INFORMATION

For the mutual information I(Z;V), we have:

$$
\begin{aligned}
I(Z;V) &= \sum_{v,z} p(v,z) \log \frac{p(v,z)}{p(v)p(z)} \\
&= \sum_{v,z} p(v,z) \log \frac{p(z|v)}{p(z)} \\
&= \sum_{v,z} p(v,z) \log p(z|v) - \sum_{v,z} p(v,z) \log p(z)
\end{aligned}
\tag{19}
$$

By substituting p(z) with the prior distribution of z, denoted as q(z), we obtain:

$$
\sum_{v,z} p(v,z) \log p(z) \geq \sum_{v,z} p(v,z) \log q(z)
\tag{20}
$$

Thus, we can derive the upper bound of mutual information:

$$
\begin{aligned}
I(Z;V) &\leq \sum_{v,z} p(v,z) \log p(z|v) - \sum_{v,z} p(v,z) \log q(z) \\
&= \sum_{v} p(v) D_{KL}[p(z|v)||q(z)] \\
&= \mathbb{E}_{p(v)} D_{KL}[p(z|v)||q(z)]
\end{aligned}
\tag{21}
$$

## B    SELECTION OF ACTIVATION FUNCTION

In our method, we applied the same softmax function to the original random matrix $A$ as used in the attention mechanism, but the sigmoid function can also be chosen here. The sigmoid and softmax functions have the same range, but softmax constrains the sum of each row in the attention.

We have tested both functions, and the following figure shows the results of the second head in the last layer in the ASR task, with hyperparameters set as $beta = 10$, $lr = 0.1$, and trained for 200 steps. The results obtained are as follows:

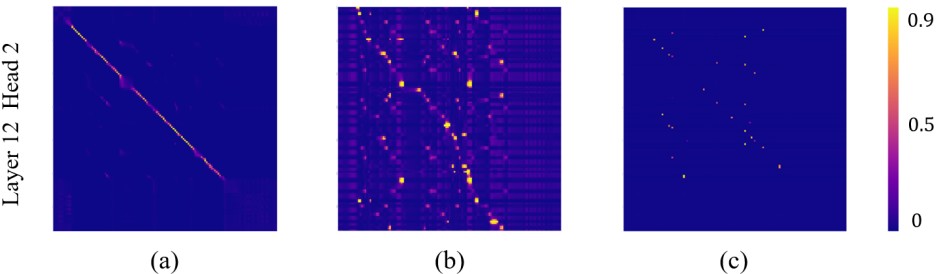

Figure 7: Effect of different activation functions on HIB.

The far left represents the original Attention result. The middle represents $A'_{HIB}$ obtained using the sigmoid function, and the right represents $A_{HIB}$ obtained using the softmax function.

In our ASR task, we chose CTC-loss. The decoding steps of the two figures can align well when using the softmax function, which is the same as the one used in the attention mechanism. The overall effective range of Fig.7(a) and Fig.7(c) is not significantly different, and they can accurately reflect the silent segments at the beginning of the input speech sequence.

In contrast, for $A'_{HIB}$ obtained using the sigmoid function, due to the lack of horizontal constraints, it tends to output at the beginning of the decoding sequence, directly overlooking the silent segments of the speech. As can be seen from Fig.7(b), its overall decoding process seems to be vertically elongated compared to normal Attention, and therefore, its decoding steps cannot align with normal Attention. Hence, we ultimately chose the softmax function.

## C  HYPERPARAMETERS

| Parameter | ASR | SID | KS | IC | Search space |
|---|---|---|---|---|---|
| Optimizer | | Adam | | | |
| Learning Rate | | 0.1 | | | {0.001, 0.01, 0.1, 1, 10} |
| Balance Factor | 10 | 0.1 | 1 | 1 | {0.01, 0.1, 1, 10, 100, 1000} |
| Noise Intensity | | 0.9 | | | {0.1, 0.5, 0.9, 0.99, 1, 2 } |
| Training Steps | 100 | 200 | 200 | 100 | {100, 200, 500, 1000} |

Table 1: HIB hyperparameters in different speech tasks

We have provided the hyperparameters of HIB for different tasks in Table 1.

