# OpenReview forum: "Head Information Bottleneck: An Evaluation Method for Transformer Head Contributions in Speech Task"
_ICLR.cc/2024/Conference — ICLR 2024 Conference Withdrawn Submission_

### Official Review · Reviewer_QeFk · 2023-10-22

**Soundness:** 2 fair
**Presentation:** 2 fair
**Contribution:** 2 fair
**Rating:** 3
**Confidence:** 3

**Summary:**

This paper proposes a novel approach for analyzing the contribution of each attention head in Transformer-based speech processing models. Leveraging information bottleneck (IB) theory, which facilitates the interpretable analysis of neural network behavior by constraining mutual information, the authors apply this method to the self-attention mechanism. Specifically, the attention weight is replaced with a trainable weight matrix so that the mutual information between V and Z (V times A) can be controlled. This proposed Head Information Bottleneck (HIB) analysis allows for the computation of the importance of each attention head. By pruning less important heads based on their computed scores, the paper demonstrates a reduction in performance degradation across various speech-related tasks.

**Strengths:**

* This is the first attempt to apply IB theory to self-attention analysis. The proposed method systematically seeks to uncover the importance of self-attention in speech processing, which could help in understanding the role of self-attention.
* The proposed method naturally aligns with IB theory. Replacing attention weight and restricting the information flow between V and Z seems to be a good approach to managing (limiting) the information flow within the self-attention module.
* Experimental results exhibit good pruning resilience that surpasses the naïve attention probability-based pruning and matches the performance of the previous AttAttr method. This empirically shows the potential utility of the proposed HIB as a useful tool.

**Weaknesses:**

* While the paper references many prior works on attention analysis, it lacks comprehensive comparative analysis. There is only one comparison with the previous work (AttAttr), and it is conducted in only a single experiment.
* The advantage of HIB requires further clarification. There are several visualizations (Figures 3 and 4) from HIB, but there is a lack of explanation/connection between these visualizations and the actual speech data. In other words, there is no intuitive demonstration of ‘what’ this attention head is focusing on. The current version of the paper is not enough to persuade readers that HIB is successfully identifying the important heads that capture important information from the input speech.
* I have several concerns about the assumptions in the paper. The paper may need more clear explanations/justifications on these points to persuade readers.
    1. First, why Z and V are assumed to be normal distributions scaled to standard normal distributions, and their covariance matrices are assumed to be identity matrices?
    2. Second, Z is a multiplication of V and Softmax(A), but the Softmax distribution is not expected to be a normal distribution; why Z is assumed to be a normal distribution?
    3. Third and most important, why does the higher similarity between the actual attention matrix and the learned HIB attention matrix indicate the greater importance of that head?

**Questions:**

* In Figure 3, why does increasing the beta ‘change’ the important attention weight locations? As I understand, important parts identified from the harsh condition (more limited information flow) should be preserved when the condition is relaxed (allowing more flow). However, it seems that important parts keep changing as beta changes.
* What do you mean by “accuracy of ASR task” (Figure 5)? The common metric for ASR is word error rate (WER), but I am confused if the used ‘accuracy’ metric is related to WER.
* If HIB yields similar results to AttAttr, what would be the advantage of HIB? In other words, why should researchers choose HIB instead of AttAttr for attribution analysis?

---

> ### Author Response · Authors · 2023-11-17
> **Response to Reviewe QeFk**
>
> Thank you for your supportive feedback on our work. We have modified the original manuscript in response to weaknesses 1，3. You may refer to the global response provided earlier and the content highlighted in blue within the article for the specific changes made. We have also taken into consideration the suggestions you provided. Below is our reply to the remaining issues:
>
> > The advantage of HIB requires further clarification. There are several visualizations (Figures 3 and 4) from HIB, but there is a lack of explanation/connection between these visualizations and the actual speech data. In other words, there is no intuitive demonstration of ‘what’ this attention head is focusing on. The current version of the paper is not enough to persuade readers that HIB is successfully identifying the important heads that capture important information from the input speech.
>
> The objective of our paper is to use HIB to identify attention heads that are beneficial for speech tasks. In future work, we will further explore the application of this method in various aspects of speech processing.
>
> > In Figure 3, why does increasing the beta ‘change’ the important attention weight locations? As I understand, important parts identified from the harsh condition (more limited information flow) should be preserved when the condition is relaxed (allowing more flow). However, it seems that important parts keep changing as beta changes.
>
> In Figure 3, as $\beta$ gradually increases, the highlighted regions gradually reduce and remain in fixed positions. It's only when $\beta$ becomes 1000 that we observe a different outcome. At this point, due to the excessively large value of $\beta$ it disrupts the normal training process, rendering the results unusable. Similar observations were made in Schulz's paper [1].
>
> [1] Karl Schulz, Leon Sixt, Federico Tombari, and Tim Landgraf. Restricting the flow: Information bottlenecks for attribution. arXiv preprint arXiv:2001.00396, 2020.
>
> > What do you mean by “accuracy of ASR task” (Figure 5)? The common metric for ASR is word error rate (WER), but I am confused if the used ‘accuracy’ metric is related to WER.
>
> The evaluation metric we employed is the accuracy calculated as 1 minus the Word Error Rate (WER).
>
> > If HIB yields similar results to AttAttr, what would be the advantage of HIB? In other words, why should researchers choose HIB instead of AttAttr for attribution analysis?
>
> Our innovation lies in introducing a novel method for interpreting the importance of attention heads, a technique we believe holds great promise for future applications. Compared to AttAttr, our method offers more convenience in multi-label task attribution. For instance, AttAttr requires attribution calculations at every position across the entire sequence length in ASR tasks, whereas HIB simplifies this by requiring only a single calculation to produce results.

---

> ### Comment · Area_Chair_zgpW · 2023-11-20
>
> Hello, reviewer. Please review the author's response. You are concerned about the assumptions in the paper. Do you think the author's response addresses your concern?

---

### Official Review · Reviewer_sbP7 · 2023-10-30

**Soundness:** 2 fair
**Presentation:** 1 poor
**Contribution:** 2 fair
**Rating:** 3
**Confidence:** 4

**Summary:**

This paper presents an information-bottleneck based approach to measure the importance of attention heads in transformer models for speech processing. The paper shows that using this approach, it is possible to prune 40% of attention heads without sacrificing much accuracy.

**Strengths:**

This paper is the first work in the literature that leverages the information bottleneck framework to understand the attention heads in transformer models for speech processing. The information theory perspective to explain the attention mechanism in transformer models is novel and interesting.

**Weaknesses:**

There are room for improvement in the presentation of this work:
1. many symbols are not defined before used and they are not easy to interpret. For example,
   -  it is unclear to the reader what"KS" is referring to in abstract;
   - symbol L and d in Eq (6) is never defined;
   - the inline equation under Eq(8) is never defined -- it looks like it is a typo ?
   - I am not sure \beta in Eq(5) and Eq(15) is the same symbol or not ? it looks like to me they should be different symbols; if not, how Eq(5) is transferred to Eq (15) ?
   -  Figure 2 is presented without any sentence mentioning what exactly it means
   -  In section 4.2, the author mentions "the impact on the dataset is 0.51% and 0.01%". How the impact is measured here? Are 0.51% and 0.01% relative WERs or absolute WERs or something else ?
   - Eq (16) `attrh` and `E_x` are not defined and very hard to understand.

2. In section 3, the author claims that they uses a variational approximation to estimate the mutual information between Z and V; then quickly after Eq(10), they stated that "according to the assumption that Z, V are both d-dimension normal distribution scaled to standard normal distribution, the mean vectors are zero and covariance are identity". First of all, V is R^{Lxd} matrix not a vector; second, where this assumption is made ? Third, is it reasonable to assume V is zero-mean, identity covariance ? This sounds a very strong assumption which does not seem to be reasonable to me.

**Questions:**

It is unclear to me what the proposed model look like ? If the attention map is replaced by a learned attention matrix, does it mean that during inference, the contribution of each of the L frames to other frame are completely fixed ? Does it mean that V is always combined using a fixed weight and how this can explain the dynamics of attention heads (when the input changes, the original attention will combine V differently, while this model is always using the fixed weight, regardless of the input) ?

---

> ### Author Response · Authors · 2023-11-17
> **Response to Reviewe sbP7**
>
> Thank you for your supportive feedback on our work. We have modified the original manuscript in response to weaknesses 1-3, 5-6. You may refer to the global response provided earlier and the content highlighted in blue within the article for the specific changes made. We have also taken into consideration the suggestions you provided. Below is our reply to the remaining issues:
>
> > I am not sure \beta in Eq(5) and Eq(15) is the same symbol or not ? it looks like to me they should be different symbols; if not, how Eq(5) is transferred to Eq (15) ?
>
> Due to rendering variations, \beta may appear differently throughout the document; however, they signify the same parameter.
>
> >In section 3, the author claims that they uses a variational approximation to estimate the mutual information between Z and V; then quickly after Eq(10), they stated that "according to the assumption that Z, V are both d-dimension normal distribution scaled to standard normal distribution, the mean vectors are zero and covariance are identity". First of all, V is R^{Lxd} matrix not a vector; second, where this assumption is made ? Third, is it reasonable to assume V is zero-mean, identity covariance ? This sounds a very strong assumption which does not seem to be reasonable to me.
>
> We have revised Chapter 3 to enhance the reading experience. Here, we adopt the same assumption as Schulz [1], Alemi [2], and Klambauer [3], presuming that $Q(Z)$ follows a standard normal distribution to simplify the computational process.
>
> [1] Karl Schulz, Leon Sixt, Federico Tombari, and Tim Landgraf. Restricting the flow: Information
> bottlenecks for attribution. arXiv preprint arXiv:2001.00396, 2020
>
> [2] Alexander A Alemi, Ian Fischer, Joshua V Dillon, and Kevin Murphy. Deep variational information
> bottleneck. arXiv preprint arXiv:1612.00410, 2016.
>
> [3] G ̈unter Klambauer, Thomas Unterthiner, Andreas Mayr, and Sepp Hochreiter. Self-normalizing
> neural networks. Advances in neural information processing systems, 30, 2017.
>
> > It is unclear to me what the proposed model look like ? If the attention map is replaced by a learned attention matrix, does it mean that during inference, the contribution of each of the L frames to other frame are completely fixed ? Does it mean that V is always combined using a fixed weight and how this can explain the dynamics of attention heads (when the input changes, the original attention will combine V differently, while this model is always using the fixed weight, regardless of the input) ?
>
> Our model is designed for attribution explanations and does not have an inference stage. We perform attribution explanations for one sample at a time, generating contribution values \(C_h\) for each head. The final results are obtained by averaging these values across the entire test dataset.

---

### Official Review · Reviewer_zTHX · 2023-11-02

**Soundness:** 3 good
**Presentation:** 3 good
**Contribution:** 3 good
**Rating:** 8
**Confidence:** 4

**Summary:**

* The paper presents a novel method to estimate the contribution of each attention head in a Transformer model. It uses the concept of information bottleneck that learns a trainable weight $A_r$ to maximize the mutual information between the random variable $Z$ and the label $Y$ while minimizing the mutual information between $Z$ and the input $X$. The similarity between the resulting head information bottleneck matrix $A_{HIB}$ and the original attention matrix $A_o$ can be used as indicator about the contributions of the attention heads.

* The proposed method is the first attempt to use information bottleneck theory to analyze the roles of attention heads.

* The experiment results on wav2vec 2.0 indicate the effectiveness of HIB on analyzing the contribution of attention heads and pruning attention heads on a few downstream tasks.

**Strengths:**

* The paper proposes a novel information bottleneck acquisition method of the attention mechanism of Transformers, which is the first attempt to use information bottleneck theory to analyze the roles of attention heads.

* The mathematical analysis of the loss function is solid and well organized, which makes the proposed method more convincing and persuasive.

* Extensive experiments on wav2vec 2.0 have been conducted to evaluate the effectiveness of HIB on analyzing the contribution of attention heads and pruning attention heads on a few downstream tasks.

* The paper conducts rigorous experiments on hyper-parameters, comparison with other methods and especially two pruning models, which demonstrate the effectiveness of the proposed method.

* The code of the method is available, making the experiments and results can be easily reproduced.

**Weaknesses:**

* In Section 3, it is assumed that the dimensions in $Z, V, Q(Z)$ are normally and independently distributed. The authors attempt to justify the reasonableness of such assumption with reference to (Klambauer et al., 2017). However, such assumptions may not be appropriate for real-world data. And the reference (Klambauer et al., 2017) is actually missing important paper title information.

* The results in Figure 5 demonstrate that AttAttr performs similar or better than the proposed method across various numbers of pruned heads. The actually suggests that the proposed method does not consistently outperforms AttAttr to a significant degree. It would be appreciated if the authors could provide more insights and analysis on this.

* The results in Figure 6 indicate that while the performance of the proposed method is satisfactory for the KS task, there is a steep decline in performance even with the pruning of just 2 or 0.1 heads for the IC and SID tasks. However, the paper lacks detailed analysis and explanations regarding the reasons behind this poor performance on the IC and SID tasks.

**Questions:**

* The presence of identical letters used to represent different terms in equations can indeed lead to confusion and misleading interpretations. For example, in Equation 5, the term “$I$” represents the mutual information, whereas in Equation 6, the term “$I$” denotes the unitary matrix.

* In the 6-7th lines of the paragraph under Figure 4, if it refers to “Layer 12 Head 3” in the Figure, and there is mention of the “fist head”, then it is likely a typo. The “first head” should indeed be corrected to the “third head” to accurately correspond with the specific layer and head being discussed in Figure 4.

* The legend entry for Figure 6 should provide clear and concise explanations of the terms “attention head KS/IC/SID” and “HIB KS/IC/SID” to avoid confusion and unclearness.

---

> ### Author Response · Authors · 2023-11-17
> **Response to Reviewe zTHX**
>
> Thank you for your supportive feedback on our work. In response to your queries, we have corrected the erroneous citations in Chapter 3 and clarified the sections of the formula that may confuse. Additionally, we have included comparative experiments and relevant analyses in Section 4 of Chapter 4. We have also revised the legends in Figure 6 to enhance readability.
>
> Our innovation lies in introducing a novel method for interpreting the importance of attention heads, a technique we believe holds great promise for future applications. Compared to AttAttr, our method offers more convenience in multi-label task attribution. For instance, AttAttr requires attribution calculations at every position across the entire sequence length in ASR tasks, whereas HIB simplifies this by requiring only a single calculation to produce results.
>
> We hope these answers clarify your question and are open to further discussion. Thank you again for your constructive feedback.

---

> > ### Comment · Reviewer_zTHX · 2023-11-23
> > **Thank authors for the response**
> >
> > Thank the authors for the response. I have also read the other reviewers' comments and authors' responses. I would keep my original rating.

---

### Official Review · Reviewer_oLGF · 2023-11-03

**Soundness:** 2 fair
**Presentation:** 2 fair
**Contribution:** 2 fair
**Rating:** 3
**Confidence:** 3

**Summary:**

This paper proposes to use the information bottleneck approach to investigate the importance of an attention head. Compared with the previous studies based on gradient analysis, the method attaches the latent variable with learnable parameters with an existing transformer-based pre-trained model to design an additional loss. After training, the method can calculate the relationship between the normal attention map and the attached attention map for each head and compute the importance based on them. The effectiveness of the proposed method was evaluated based on widely-used w2v2-based Librispeech ASR experiments or other speech processing downstream tasks.

**Strengths:**

- Besides the layer analysis, attention head analysis in the transformer architecture is fundamental. There is not so much work conducting the head-level analysis, especially in speech processing
- The method is theoretically founded based on the information bottleneck.
- The effectiveness is evaluated by various speech processing downstream tasks (e.g., ASR, SID, KS, and IC).
- Good reproducibility (they use the public data and will release the source code).

**Weaknesses:**

- Clarity: The paper does not fully explain the information bottleneck concept. For example, equation (5) introduces the core objective of this theory, but there is little information about why this objective function yields the information bottleneck and what $Z$ exactly means. I recommend you carefully explain this part more conceptually.
- Clarity: There are missing explanations for several variables introduced in the equation, e.g., $L$, $T$, $d$, etc. Especially, I'm confused $L$ and $T$ (also $a_i$ and $a ^r _i$). Are they the same or different? Please carefully explain all variables.
- Clarity: Generally, the experimental discussions miss the detailed configurations (see my questions), making it difficult to understand the analyses.
- Clarity: All figures are too small.
- Novelty and effectiveness: I could not fully find the benefits of this method compared with the conventional method, e.g., AttAttr. The performance difference is minimal. Also, although the proposed method claims that it only requires forward computation compared with AttAttr, according to the discussion in Section 3,  it seems to train the model based on the additional loss and parameters and requires backpropagation. Thus, I could not find the benefit of the proposed method.

**Questions:**

- I'm a bit confused. Section 3 says that "this method operates during the forward propagation process of an already-trained network." However, it is based on loss optimization, and we still need to perform backpropagation (not for all parameters but parameters related to the information bottleneck) to realize this method. Can you clarify this part?
- I could not fully understand what is actually trained. I could read that attention matrix $A$ is trained in the final paragraph in Section 3. However, this matrix will be variable depending on the input length, and it is difficult to train. Can you clarify it?
- Can you clarify the experimental condition of Figure 2?
- Which head did you use for Figure 3?
- Section 4.2: "When these two heads are respectively masked, the impact on the dataset is 0.51% and 0.01%, representing heads with relatively minor and major impacts on the ASR task, respectively." Is it about Figure 2?
- Eq. (16) needs more explanations. I could not understand what this value means.
- Section 4.3 "We randomly extracted 200 samples from the Librispeech train-100
dataset for training." Do 200 samples mean 200 utterances? Did you only use 200 utterances to train the model? Can you clarify it?
- Section 4.3: How to calculate the accuracy? In ASR, it is not easy to calculate a simple accuracy, and people use the word error rate (edit distance between the reference and hypotheses) to evaluate the model performance.
- Why didn't you compare the proposed method with AttAttr in Section 4.4?
- Section 5: What is "attention score?"
- Can you apply this method to the cross attention? The cross attention would often be more important to interpret the relationship between input and output in an encoder-decoder architecture. If this method is applied to the cross attention, it would provide interpretability in such an architecture.


Other suggestions
- in Abstract: KS task --> keyword spotting task. Please avoid using the abbreviation without a definition in the abstract. Also, it is better to add which KS task you are using.
- In Introduction: Whisper is not an SSL model
- In Introduction: (Finke et al., 1997; Stolcke et al., 2000) are not representative studies for GMM-HMM. You may refer to Rabiner, Lawrence R. "A tutorial on hidden Markov models and selected applications in speech recognition." Proceedings of the IEEE 77.2 (1989): 257-286 or other representative papers.
- Section 2.2: I recommend you emphasize that AttAttr (Hao et al., 2021) would be used for your main comparisons here.
- I think this paper would be more attractive if it is applied to other applications (e.g., LLM, Vision transformer) additionally. Please consider the additional evaluations.

---

> ### Author Response · Authors · 2023-11-17
> **Response to Reviewer oLGF**
>
> Thank you for your supportive feedback on our work. We have modified the original manuscript in response to questions 3-6,9, and 10. You may refer to the global response provided earlier and the content highlighted in blue within the article for the specific changes made. We have also taken into consideration the suggestions you provided. Below is our reply to the remaining issues:
>
> >I'm a bit confused. Section 3 says that "this method operates during the forward propagation process of an already-trained network." However, it is based on loss optimization, and we still need to perform backpropagation (not for all parameters but parameters related to the information bottleneck) to realize this method. Can you clarify this part?
>
> Your understanding is correct, and we apologize for any confusion caused by our previous statements. We aim to clarify that our attribution method is distinct from those that require the introduction of new structures during model training. Our technique is designed to be applied directly to models that have already been trained. This is a significant advantage over other attention head-based attribution methods that necessitate retraining from scratch and are unable to evaluate pre-trained models. Consequently, our approach substantially reduces the cost required to assess models.
>
> >I could not fully understand what is actually trained. I could read that attention matrix  $A$ is trained in the final paragraph in Section 3. However, this matrix will be variable depending on the input length, and it is difficult to train. Can you clarify it?
>
> Our method is an attribution technique specifically tailored for variable-length speech signals, which necessitates conducting the attribution training on one sentence at a time.
>
> >Section 4.3 "We randomly extracted 200 samples from the Librispeech train-100 dataset for training." Do 200 samples mean 200 utterances? Did you only use 200 utterances to train the model? Can you clarify it?
>
> Our model is an attribution model; therefore, it requires attributing each sentence individually. Considering time costs, we extracted 200 sentences from the test set for attribution testing.
>
> >Section 4.3: How to calculate the accuracy? In ASR, it is not easy to calculate a simple accuracy, and people use the word error rate (edit distance between the reference and hypotheses) to evaluate the model performance.
>
> The evaluation metric we employed is the accuracy calculated as 1 minus the Word Error Rate (WER).
>
> >Can you apply this method to the cross attention? The cross attention would often be more important to interpret the relationship between input and output in an encoder-decoder architecture. If this method is applied to the cross attention, it would provide interpretability in such an architecture.
>
> Our method is applicable to any multi-head attention mechanism, including cross-attention. However, in this paper, we have focused our research on self-attention-based pre-trained speech models. In the future, we plan to explore the application of this method across a broader range of fields.
>
> We hope these answers clarify your question and are open to further discussion. Thank you again for your constructive feedback.

---

> ### Comment · Area_Chair_zgpW · 2023-11-20
>
> Hello, reviewer. Please review the author's response. You are concerned about the benefits of this method compared with the conventional method. Do you think the author's response addresses your concern?

---

### Author Response · Authors · 2023-11-17
**Global Response**

We express our profound gratitude to each reviewer for dedicating their time to scrutinize our paper and for offering constructive recommendations that have significantly contributed to its refinement.

In response to the feedback received, we have undertaken substantial revisions to improve the manuscript's readability and clarity. The principal amendments are highlighted below, and we have also indicated these changes in blue within the document for easier identification:
- An introduction to the Information Bottleneck theory has been added in Chapter 2 to provide a foundational understanding.
- In Chapter 3, we have revised the presentation of the equations, standardizing and defining previously unspecified symbols for consistency and clarity.
- A detailed explanation of the contribution metric $C_h$ is now included in Chapter 3, enhancing the paper's comprehensibility.
- At the beginning of Chapter 4, we have elaborated on Figure 2, offering a more comprehensive description.
- Section 3 of Chapter 4 includes a detailed explanation of the $I_h$ formula (previously Equation 16 in the original manuscript).
- In Section 4 of Chapter 4, we have added comparative experiments with AttAttr and provided an analysis of the results.
- We have corrected various errors throughout the manuscript to improve its overall readability further.

We want to reiterate our approach: we have designed an attribution method for pre-trained models that can be applied to any multi-head attention mechanism. This method is attributed on a per-sample basis, resulting in the contribution score $C_h$ for each sample. The experimental results presented in Chapter 4 reflect the average across multiple samples.

For other concerns raised, we will address each reviewer’s comments individually in separate responses.